

# 1 Interactive coupling of a Greenland ice sheet model in NorESM2

Heiko Goelzer[1], Petra M. Langebroek[1], Andreas Born[2], Stefan Hofer[3,4], Konstanze Haubner[2], Michele
Petrini[1], Gunter Leguy[5], William H. Lipscomb[5], Katherine Thayer-Calder[5]
[1]NORCE Norwegian Research Centre, Bjerknes Centre for Climate Research, Bergen, Norway
[2]Department of Earth Science, University of Bergen, Bjerknes Centre for Climate Research, Bergen, Norway
[3]School of Geographical Sciences, University of Bristol, Bristol, UK
[4]Department of Geosciences, University of Oslo, Oslo, Norway
[5]Climate and Global Dynamics Laboratory, NSF National Center for Atmospheric Research, Boulder, CO, USA
*Correspondence to*: Heiko Goelzer ([heig@norceresearch.no](mailto:heig@norceresearch.no))
**Abstract**
On the backdrop of observed accelerating ice sheet mass loss over the last few decades, there is growing
interest in the role of ice sheet changes in global climate projections. In this regard, we have coupled the
Norwegian Earth System Model (NorESM) with the Community Ice Sheet Model (CISM) and have
produced an initial set of climate projections including an interactive coupling with a dynamic Greenland
ice sheet. Our focus in this manuscript is the description of the coupling, the model setup and the
initialisation procedure. To illustrate the effect of the coupling, we have further performed one chain of
experiments under historical forcing and subsequently under high future greenhouse gas forcing (SSP5-
8.5) until 2100 and extended until 2300. We find a limited impact of the dynamical ice sheet changes on
the global response of the coupled model under the given forcing and experimental setup when comparing
to a standard CMIP6 simulation of NorESM with a fixed ice sheet.

## 22 1 Introduction

Ice sheets are regularly discussed and studied in the context of their future sea-level contribution (Seroussi
et al., 2020; 2024; Goelzer et al., 2020) and as potential tipping elements in the Earth system (e.g., Pattyn
et al., 2018). However, ice sheets are recognised not only as Earth system components that strongly
respond to climate changes, but also for their potential to influence climate in turn through interactions
with atmosphere, land and ocean (e.g. Vizcaino, 2014). Studying ice sheet - climate interactions therefore
requires the ice sheets to be coupled to the other Earth system components. These feedbacks become
relevant on long enough timescales, typically centennial to multi-millennial. Relevant large-scale
processes that give rise to feedbacks include the influence of a changing ice sheet topography on surface
temperature and atmospheric circulation (Merz et al., 2014; 2016), changes in runoff and iceberg fluxes
that modify ocean stratification (Martin & Biastoch, 2023) and circulation, and ice sheet expansion or
retreat that change the planetary albedo and the potential for vegetation, modifying the radiation and
surface energy budget (Vizcaino et al., 2010; Stone and Lunt, 2013).



Given the long timescales on which some of these interactions manifest, modelling climate–ice sheet
interactions has until recently been mostly out of reach for high-complexity, high-resolution Coupled
Model Intercomparison Project (CMIP) models, with CESM2 and UKESM being the only models that
delivered coupled climate-ice sheet simulation results under CMIP6 (Muntjewerf et al., 2021; Smith et
al., 2021). A large body of work has also focussed on models of lower complexity and/or lower resolution
to advance coupled climate–ice sheet science over the last two decades (e.g., Huybrechts et al., 2002;
Ridley et al., 2005; Mikolajewicz et al., 2007; Ganopolski et al., 2010; Goelzer et al., 2011; Roche et al.,
2014; Gregory et al., 2020). The challenge inherent in these simulations from the ice sheet perspective is
bridging the gap between climate boundary conditions produced at a spatial resolution of up to several
degrees to the finer ice sheet scale (typical resolution of only a few km). In addition, climate biases often
translate into biases in ice sheet state, which has been mitigated e.g. by use of anomaly methods or ad-
hoc corrections (e.g. Goelzer et al., 2012). While these problems are typically reduced with higher
resolution and lower biases, they remain some of the most important challenges when implementing ice
sheet dynamics in climate models. A key advance, paving the way to include ice sheets eventually in
CMIP-type climate models, was the advent of efficient downscaling procedures (Vizcaino et al., 2010;
2013; 2014; Sellevold et al., 2019), that produce relatively high-quality surface mass balance (SMB) as
ice sheet forcing. These exploit a strong elevation (temperature) dependence of some surface mass and
energy balance components, in particular of the melt process, which is why they were first successfully
implemented for simulations including the Greenland ice sheet (GrIS). For the significantly colder
Antarctic ice sheet at present, the SMB is dominated by the distribution of snowfall, which is notoriously
difficult to downscale and hinges on the native resolution of atmospheric dynamics. Another remaining
challenge for coupled modelling are how to treat the interaction of ice sheets and ocean for the narrow
fjords of Greenland and the ice shelves in Antarctica, that are equally not resolved in global climate
models. Furthermore, initialising the climate-ice sheet system is a difficult task due to the specific
response timescales of the different systems. There is a strong interest of many modelling groups
worldwide to overcome these challenges and to work towards coupled climate–ice sheet simulations
leading up to CMIP7. These coupled simulations are supported by a community effort under the Ice Sheet
Model Intercomparison Project (ISMIP7).
In this paper, we describe the implementation and first results of GrIS coupling in the Norwegian Earth
System Model (NorESM), which builds on a similar development for CESM2 and the Community Ice
Sheet Model (CISM, Lipscomb et al., 2019). We describe the model with focus on climate–ice sheet
interactions and initialisation (Sect. 2) and the experimental setup (Sect. 3). We show results in section 4
and close with Discussions (Sect. 5) and Conclusions (Sect. 6).
**2 Model description**
In this section, we describe our novel coupled modelling framework consisting of climate and ice sheet
components, the dynamic coupling and the initialisation procedure.



## 2.1 The Norwegian Earth System model (NorESM)

NorESM is a full-complexity CMIP-type Earth system model (ESM) mainly developed by the Norwegian Climate Centre (NCC) consortium. Here, we discuss the model version NorESM2 (Seland et al., 2020), which contributed to CMIP6 (Eyring et al., 2016) without dynamic ice sheets (NorESM2fixed). We have expanded from this CMIP6 version and included interactive coupling with a dynamic GrIS component (Sect. 2.2). NorESM2 shares many technical features with CESM2 (Danabasoglu et al., 2020) because the fundamental model components for land (CLM), atmosphere (CAM), sea ice (CICE), and land ice (CISM) are the same (Fig.1). The coupling interface between the ice sheet on the one hand and atmosphere and land models on the other hand is also inherited from CESM2, using the same elevation-class approach (Sec 2.3) to provide surface mass and energy balance from the atmosphere (CAM) via the land model (CLM) to the ice sheet model. The ocean model in NorESM2 (BLOM), the ocean biogeochemical component (iHAMOCC) and extended atmospheric chemistry options (in CAM) are distinguishing features and lead to a different climate sensitivity compared to CESM2— specifically, a lower transient climate response (Seland et al., 2020). In version 2, NorESM can run with only one interactive ice sheet domain at a time (here Greenland). Implementing an Antarctic ice sheet and paleo ice sheets are subject to future model development.

We have run coupled climate-ice sheet simulations with NorESM2 at two different horizontal resolutions of the atmosphere model, called NorESM2-MM (1° x 1°) and NorESM2-LM (2° x 2° resolution) that have both uncoupled (NorESM2fixed) contributions to CMIP6 to compare to. In the following we focus mainly on the higher resolution version and use the name NorESM2 for NorESM2-MM unless indicated otherwise.

## 2.2 The Community Ice Sheet Model (CISM)

CISM is a thermodynamically-coupled ice sheet model (Lipscomb et al., 2019), run on a structured grid, that can be used for both coupled (Muntjewerf et al., 2020a; b; Petrini et al., 2024) and standalone applications (Lipscomb et al., 2021; Berdahl et al., 2023; Rahlves et al., 2024).

As the GrIS component in NorESM2, we use CISM at 4x4 km horizontal resolution and with 11 unequally spaced vertical levels on a variable-thickness sigma coordinate. The ice sheet domain is laid out on a standard polar stereographic projection and restricted to the main Greenland island. The momentum balance is solved with the higher-order depth integrated viscosity approximation (DIVA) approach (Goldberg, 2011; Robinson et al., 2022) including longitudinal stress transmission in a computationally efficient vertically averaged setup. We use a basal sliding law following Schoof et al. (2005) with the option to locally calibrate basal friction coefficients (Lipscomb et al., 2021) that we exploit in the initialisation approach described in Sect. 2.4. Bedrock change due to glacial isostatic adjustment is not activated. The basic ice sheet model configuration is similar to the NSF NCAR-CISM contribution to the ISMIP6 (Nowicki et al., 2020) standalone projections (Goelzer et al., 2020).





**2.3 Coupled Climate - Ice Sheet interactions**
**Surface energy balance and surface mass balance**
In NorESM2, glacier and ice sheet surfaces are treated as an additional land surface type of the land model
CLM. This implies that the surface energy and mass balance are computed by the land model, which
passes the surface mass balance (SMB) and ice surface temperature as a forcing to CISM once a year.
The SMB is calculated as the difference between accumulation (snowfall and refreezing of rainfall and/or
previously melted snow within the snowpack) and ice loss from surface melt and sublimation:
SMB = *Snowfall + Refreezing – Melt – Sublimation.*
The available energy to melt snow and ice is calculated from the sum of net surface radiation, latent and
sensible turbulent heat fluxes, and ground heat fluxes at the atmosphere/land interface over glaciated grid
cells (Lawrence et al., 2019). The influence of elevation on both surface melt energy and SMB (Hermann
et al., 2018; Van de Wal et al., 2012) poses a challenge in bridging between the relatively low horizontal
resolution in CLM (here 1° or 2°) and the higher CISM horizontal resolution (here 4 km). This is
particularly true at the ice sheet margins, where resolving steep SMB gradients becomes difficult at coarse
resolution. CLM addresses this challenge by calculating the SMB at multiple elevation classes (ECs)
which allows to account for subgrid-scale elevation variations over glaciated land units (Lipscomb et al.,
2013; Vizcaino et al., 2014; Sellevold et al., 2019; Muntjewerf et al., 2021). To encompass the full range
of CISM grid surface elevations while adequately representing subgrid-scale topographic variations, ten
ECs are considered with boundaries at 0, 200, 400, 700, 1,000, 1,300, 1,600, 2,000, 2,500, 3,000, and
10,000 m (Muntjewerf et al., 2021, Petrini et al., 2023). The choice of this non-uniform boundary
distribution is explained by the larger number of ECs needed to capture the steep lower topography at the
ice sheet margins, as opposed to a relatively flat high-elevation terrain in the ice sheet interior (Sellevold
et al., 2019). In each EC, surface energy fluxes and their impact on SMB are calculated independently.
First, the CLM grid cell near-surface temperature (corresponding to the CLM mean grid cell elevation) is
adjusted to the 'virtual' elevation in each EC using a uniform lapse rate of -6 °K/km. The temperature in
each EC is then used to calculate EC-specific potential temperature, specific humidity, air density, and
surface pressure, assuming vertically uniform relative humidity. The CLM grid cell precipitation does not
vary through ECs but is partitioned into snow or rain based on the elevation-corrected near-surface
temperature in each EC. If the downscaled temperature is below -2°C, precipitation is assumed to be
100% snow, whereas for temperatures above 0°C it is considered as 100% rain. For intermediate
temperatures between -2 and 0 °C, a linear interpolation is applied to determine the rain-to-snow ratio
(Muntjewerf et al., 2021). Snowfall is converted to ice when the depth of the snowpack exceeds a
threshold of 10 m water equivalent, whereas for lower snowpack depth, the accumulated snow does not
directly contribute to the SMB. Liquid and solid precipitation and the EC-specific interpolated fields are
used to calculate the SMB in each EC. After this calculation, the SMB is downscaled to the higher-
resolution CISM domain through a horizontal bilinear interpolation and a linear vertical interpolation
between ECs adjacent to the CISM grid cell elevation. Following these interpolations, the discrepancy
between total mass accumulation and loss in the source (CLM) and destination (CISM) grids is calculated,





and two different normalisation factors (one for the accumulation region, and one for the ablation region)
are applied to achieve mass conservation. The CLM near-surface temperature is remapped from CLM to
CISM using the same EC method, with the only difference being that no normalisation factor is applied
after the downscaling. More details on the coupling between CLM and CISM and on the ECs methods
can be found in Muntjewerf et al. (2021) and Sellevold et al. (2019).
In the results section below, we will compare the output of this EC approach implemented in NorESM
(NorESM2-EC) over the historical period with two different results from the regional climate model MAR
v3.12. In one case the output is produced by forcing MAR with lateral boundary conditions from the
CMIP6 version of NorESM2-MM (NorESM2-MAR). Note that this version of NorESM2 does not
include an interactive ice sheet model and represents a different ensemble member with different inter-
annual and inter-decadal variability. In the other case, MAR is forced with lateral boundary conditions
coming from the reanalysis data set ERA5 (ERA5-MAR).
**Ice sheet surface topography**
To include the impact of changing ice sheet surface topography on atmospheric circulation, we adopt an
asynchronous procedure that modifies the restart files of the atmospheric model (Lofverstrom et al.,
2020). Topographic changes on the GrIS domain are interpolated and incorporated in the high-resolution
input dataset for the atmospheric component (CAM). Surface topography and surface roughness are then
re-calculated and written into the CAM restart file. The procedure is time-consuming and model progress
is paused during the update. Including the update at runtime instead would be desirable but requires
substantial recoding of the way topography and roughness boundary conditions are currently handled in
CAM. In the present experiments we update the topography every five years, in line with the restart
checkpoint frequency in our model runs and with earlier experiments with CESM2 (Muntjewerf et al.,
69 2021).

**Melt and freshwater fluxes**
As described above, the ice sheet surface is treated as an additional surface type in the land model, and
surface mass and energy calculations are handled by CLM. Surface meltwater runoff is consequently also
handled by CLM and routed to the ocean through the runoff scheme (MOSART). This liquid runoff is
coupled on hourly timescales at the time resolution of the land model. Ice sheet calving fluxes (i.e., solid
ice discharge) are converted to freshwater and passed directly to the ocean, where the energy needed to
melt ice is taken from the ocean heat reservoir. Solid ice fluxes are cumulated and passed to the ocean
annually.

**Ice–ocean interactions**
Our model does not include direct effects of the ocean on the ice sheet (e.g., via ocean temperature or
salinity). Also, the ice sheet model is restricted to simulating grounded ice, with all floating ice removed
immediately. The spatial scale of narrow marine-terminating outlet glaciers around Greenland is on the





order of only a few kilometres, while a typical horizontal resolution of the ocean model is on the order of
100 km (here at 1° x 1°). Resolving their interactions is therefore challenging. Complex interactions
between the outflowing glacial meltwater, inflowing ocean water, sea-ice and icebergs and variations in
local bathymetry and glacier geometry in ~200 individual fjords complicate the situation. Feasible
approaches are currently mostly found in simple parameterisations describing the impact of the ocean on
the ice sheet (e.g., Slater et al., 2019; 2020). In the absence of dedicated oceanic forcing of the marine-
terminating outlet glaciers in our model, glaciers are simulated to respond passively to changes in inland
inflow and SMB and deliver excess mass to the ocean (e.g. Muntjewerf et al., 2020a; b).

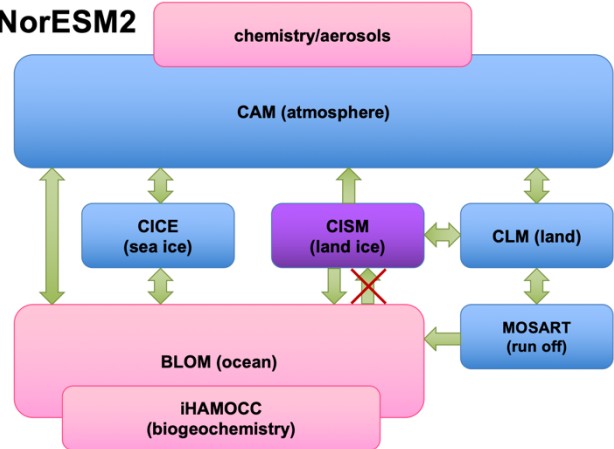

**Figure 1. NorESM2 model components**

**2.4 Initialisation**
The aim of our initialisation approach is to produce a pre-industrial coupled model configuration (for
simplicity represented by year 1850), that is close to steady state for the climate and ice sheet components.
In this first coupled setup with NorESM2, we achieve that by initialising the ice sheet as close as possible
to the observed present-day configuration, under SMB forcing derived from a pre-industrial simulation
of NorESM2 without ice sheet coupling (NorESM2fixed). The arguments for admitting this slight
inconsistency (pre-industrial forcing vs present-day ice sheet configuration) are that i) we do not know
the precise ice sheet geometry before the start of routine satellite observations in ~1990, ii) differences
between the pre-industrial and present-day ice sheet are likely small compared to what can be resolved
by the atmospheric component and iii) the climate components in NorESM2fixed have had the present-
day ice sheet geometry as topographic boundary condition in all experiments, including the pre-industrial.
Furthermore, this approach facilitates the setup and reduces the preparation time of the coupled model, as
it can be used with the tuning of an existing NorESM2fixed configuration from CMIP6.





### 2.4.1 Ice sheet model initialisation

For the coupled experiments, our method leans on our experience with standalone ice sheet simulations (e.g., Goelzer et al., 2020; Rahlves et al., 2024) and attempts to minimise the initial drift arising from introducing the ice sheet component into the global model. To that end we have calibrated the basal friction parameters (Lipscomb et al., 2019) of the ice sheet model to closely reproduce the present-day observed ice sheet elevation when forced with output from NorESM2fixed over the pre-industrial period. We also use three options implemented in CISM that control the behaviour of ice at the margins: 1) an option to remove ice caps and glaciers in the periphery that are not connected to the main ice sheet (option 'remove_ice_caps'); 2) the ice sheet is constrained by masking to the observed ice extent, allowing for retreat but not expansion of the ice sheet area beyond the present-day margins (option 'force_retreat' with constant mask); 3) Ice is not allowed to form in locations disconnected from the main ice sheet (option 'block_inception'). This means that new ice sheet cells can only form by flow from an already existing cell. In all three cases, ice thickness is set to zero and ice mass is removed as calving flux. These constraints are justified for forcing scenarios where we expect an ice sheet extent similar or retreated compared to today (historical and future periods). In other cases, e.g. glacial periods, this approach should be modified.

In combination, masking and calibration of the basal friction parameters are means to practically deal with the climatic biases in NorESM2 and the limitations of the ice sheet model. The dynamic behaviour of the model is somewhat impacted by these choices (e.g. Berends et al., 2023), but the result is an overall better agreement with the ice sheet surface elevation to which the climate model is already relaxed (Fig. 2).

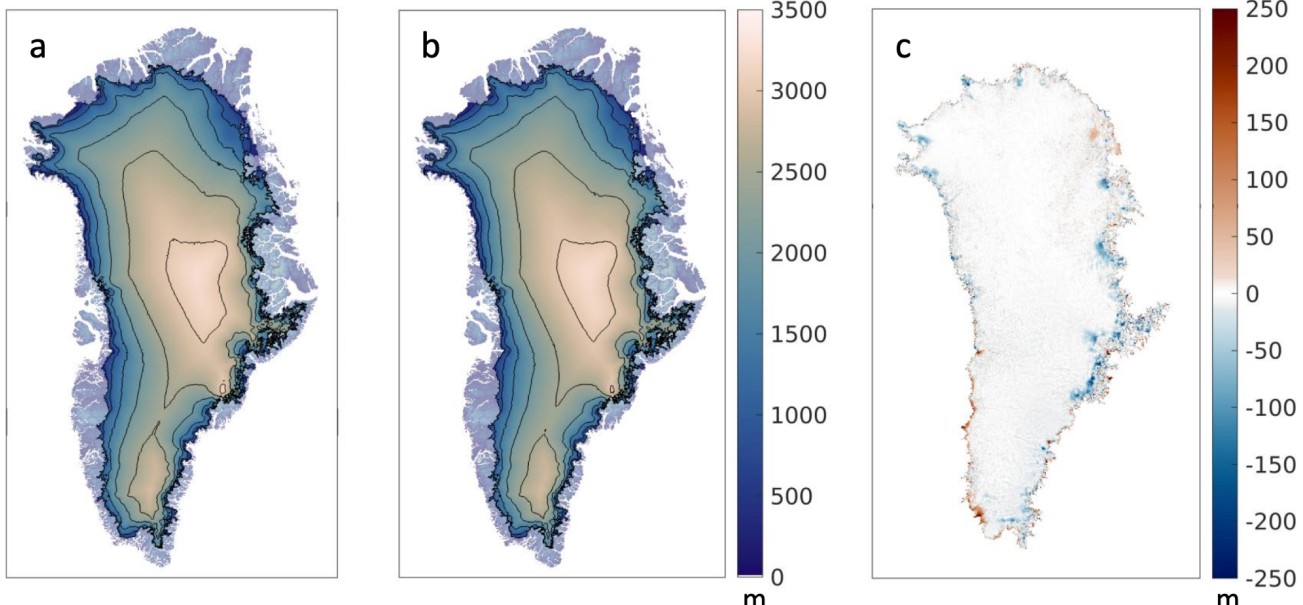

**Figure 2. Ice sheet surface elevation. a) Target surface elevation based on present-day observations. b) Ice sheet model surface elevation after initialisation for year 1850. c) Difference in surface elevation on the modelled ice mask.**





33         **2.4.2 Coupled model initialisation**

The desired consequence of the modelling decisions described in the last section is to minimise model
drift and rapidly reach a quasi-equilibrium for the coupled system with an ice sheet geometry close to
observed. It has allowed us to perform coupled simulations with very limited model drift after a short
relaxation of only 50 years (c1850 in Table 1). This is a strong benefit over other approaches that require
relatively expensive iterations to bring the ice sheet and climate states into agreement (e.g., Fyke et al.,
2014; Lofverstrom et al., 2020; Muntjewerf et al., 2020a; b). A slight increase in precipitation over
Greenland margins in response to the coupling observed during preliminary tests was further compensated
by initialising the ice sheet to a slightly biased surface mass balance forcing. Instead of calculating the
long-term mean SMB from the last 50 years of a pre-industrial steady state experiment of NorESM2fixed,
we use only the 25 years with the highest SMB for the improved initialisation. As opposed to a slight
mass gain in the preliminary forward experiment, the result is a small overall ice sheet mass loss, as the
ice sheet relaxes to the ensuing lower SMB in the forward experiment (Fig. 3d).
**3 Experimental setup**
We have performed one chain of experiments (Table 1) that follow a subset of the protocol for coupled
climate–ice sheet simulations (Nowicki et al., 2016) of the Ice Sheet Model Intercomparison Project for
CMIP6 (ISMIP6). The first coupled experiment (c1850) is a 50-year relaxation in which the climate and
ice sheet are first brought together after separate initialisation. Following is a standard historical
experiment (cHIST) from 1850-2014, and a projection under forcing scenario SSP5-8.5 to 2100
(cSSP585), that is further prolonged with a scenarioMIP extension (O'Neill et al., 2016) for SSP5-8.5 to
2300 (cSSP585Ext). We also performed a control experiment continuing the standard CMIP6 pre-
industrial experiment for 350 years (cControl). For all coupled experiments, we compare to results from
uncoupled experiments (NorESM2fixed, climate simulations indicated with "n", Table 1) to evaluate the
impact of the coupling, albeit with only one ensemble member per model setup.
**Table 1. Experiment overview.**

| Coupled experiments (c) | Uncoupled experiments (n) | Time | Comment |
|---|---|---|---|
| - | n1850 (NorESM2fixed) | 50 years | Standard CMIP6 pre-industrial experiment |
| - | ISM spinup | 5000 years | Standalone ice sheet spinup to NorESM2 SMB |
| c1850 | n1850 | 50 years | Spinup (or coupled initialisation) |
| cHIST | nHIST | 1850 – 2014 | Historical experiment |
| cSSP585 | nSSP585 | 2015 – 2100 | Projection |
| cSSP585Ext | nSSP585Ext | 2101 – 2300 | ScenarioMIP prolongation * |
| cControl | nControl | 350 years | Control experiment under preindustrial forcing |



* SSP585Ext extends SSP585 to year 2300 with $CO_2$ emissions that are reduced linearly starting in
2100 to less than 10 GtC $yr^{-1}$ in 2250 and constant during the last 50 years. Other emissions are held
constant at 2100 levels.

# 4 Results

## 4.1 Simulation over the historical period

Over the historical period, coupled and uncoupled experiments show overall a similar mean climate
evolution (Fig. 3a-c). There are differences between the phasing of their interannual and inter-decadal
variability, but this is to be expected in freely evolving (i.e., not nudged to observations) ESM simulations.
The ice sheet exhibits a small mass loss (positive sea-level contribution) of similar magnitude in the
historical experiment cHIST and the control experiment cControl (Fig. 3d), as a result of the initialisation
to slightly biased SMB forcing described above (Sect. 2.4.2). The overall mass loss rate over the historical
period is comparable to reconstructions (Zuo and Oerlemans, 1997; Box and Colgan, 2013), while
episodes of readvance and retreat suggested e.g. by Bjørk et al. (2012) are not captured.

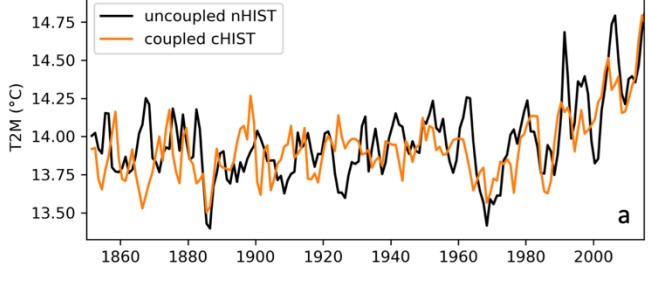
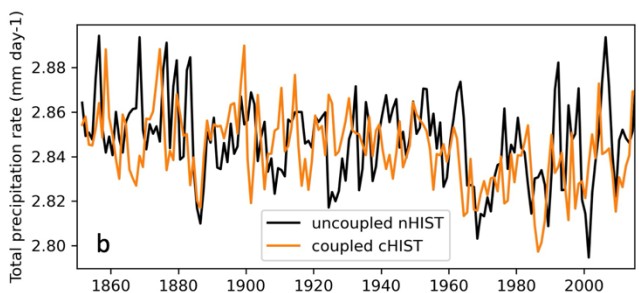
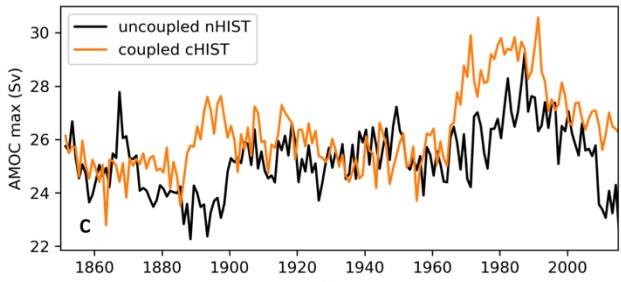
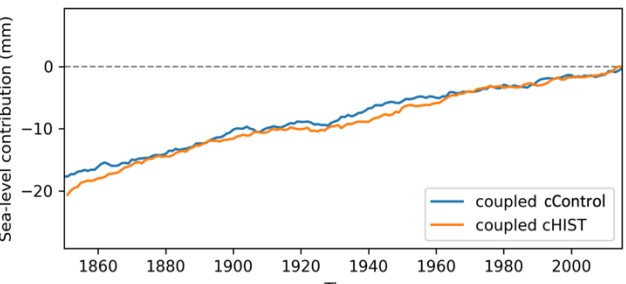

**Figure 3. Climate and ice sheet evolution over the historical period. Coupled (orange) and uncoupled (black) evolution of a) two-**
**meter air temperature (T2M), b) total precipitation rate c) Atlantic meridional overturning circulation (AMOC). d) sea-level**
**contribution from the GrIS for coupled experiments cHIST (orange) and cControl (blue).**





**4.2 SMB evaluation over the reanalysis period**

Figure 4 shows the mean SMB over the period 1960-1989 as simulated directly by NorESM2-EC (i.e., NorESM with elevation classes to downscale the SMB within the model) compared to a dynamically downscaled SMB with the regional model MAR (NorESM2-MAR, Fettweis et al., 2017). This is further compared to the SMB as obtained by MAR when forced by the ERA5-observational product for the same period (ERA5-MAR), which can be seen as our observation-based target. While NorESM2 by itself (NorESM2-EC) captures the main features (north-south gradient, high SMB in the south-east, negative SMB in the central west), the dynamically downscaled products show considerably more detail and larger areas of negative SMB around the margins. Strong similarity between the two MAR products indicates that the dynamical downscaling has a larger impact on the results than the global boundary condition (NorESM2-MAR vs ERA5-MAR).

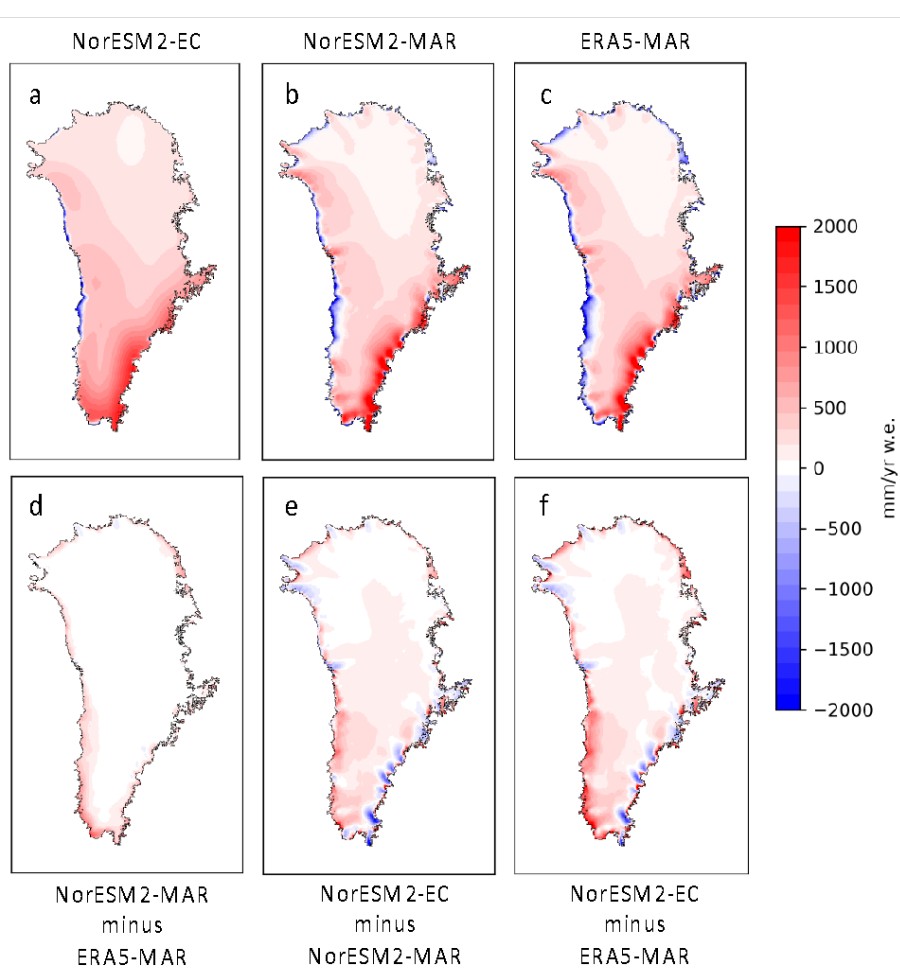

**Figure 4 Mean surface mass balance (SMB) over the period 1960-1989 from a) NorESM2-EC, b) NorESM2-MAR and c) ERA5-MAR and differences (d-f). All three fields are masked to the modelled ice sheet area in NorESM2 at the end of year 2014.**





Comparing the mean 1960-1989 SMB of NorESM2 with dynamically downscaled products NorESM2-
MAR and ERA5-MAR shows that precipitation is smoothed out into the interior in the south-east, and
topographically driven precipitation is generally not well resolved due to the relatively coarse resolution
of the atmosphere model. A comparison with NorESM2-LM with a 2° horizontal resolution in the
atmosphere illustrates these biases further (see supplementary Fig. S1). SMB around the margins is
generally too high, which can partly be explained by a cold bias of the simulated near-surface
temperatures over GrIS margins (cf. Seland et al., 2020). This is supported by the difference between
NorESM2-MAR and ERA5-MAR, indicating that even after downscaling the SMB is biased high in
NorESM2-MAR compared to the reanalysis-driven run.

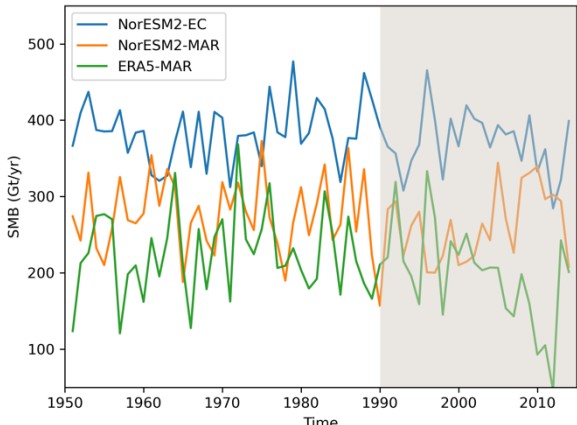

**Figure 5 Historical total surface mass balance (SMB) variations integrated over the modelled ice sheet area.**
Due to the biases described above, the spatially integrated SMB is higher in NorESM2 (380 Gt/yr)
compared to both NorESM2-MAR (284 Gt/yr) and ERA5-MAR (230 Gt/yr) (Fig. 5). Comparison
between NorESM2-EC and NorESM2-MAR shows that the NorESM2 version used for downscaling with
MAR is a different ensemble member with a different inter-annual and inter-decadal variability. This
illustrates that direct comparison on inter-annual and even multi-decadal time scales of individual
ensemble members with observations is problematic. That also applies to SMB trends after 1990 that are
negative in NorESM2-EC and seemingly of the right sign when compared with ERA5-MAR, albeit with
a muted response (-1.2 Gt/yr vs. -5.3 Gt/yr). Comparison with NorESM2-MAR with a positive SMB trend
(4.0 Gt/yr) however clearly shows that the inter-annual/ inter-decadal variability in the ESM is not aligned
with observations/reanalysis and can have considerable mismatch over these time intervals. The SMB
trends after 2000 show increasing amplitude (both positive and negative) with -6.1 (7.9) [-8.9 Gt/yr] for
NorESM2 (NorESM2-MAR) [ERA5-MAR].
The SMB variance over the period 1960-1989 in NorESM2 (1750 Gt/yr) is lower compared to NorESM2-
MAR (2185 Gt/yr) and much lower compared to ERA5-MAR (2915 Gt/yr), which we attribute to an
under-developed ablation area in NorESM2 that prohibits inter-annual temperature variations to fully
translate to variations in melt and runoff. The Greenland cold bias in NorESM2 can explain the difference
in variance between NorESM2-MAR and ERA5-MAR in a similar way.




## 4.2 Future projection

Global mean temperature increases by ~3.5 °C between 2014 and 2100 and by ~10 °C in 2300 under SSP5-8.5 and extended forcing (Fig. 6a). Northern Hemisphere sea-ice extent dramatically decreases as a result (Fig. 6b), with the minimum extent reaching zero (sea-ice free summer Arctic) by the beginning of the 22st century and a maximum extent approaching zero by the beginning of the 23rd century (practically sea-ice free Arctic year-round). The Atlantic Meridional Overturning Circulation (AMOC) shows a decline already at the end of the historical experiment, which continues over the 21st and 22nd century to a near complete shutdown state at the end of the 23rd century (Fig. 6c).

Most global climate characteristics show similar behaviour in the coupled and uncoupled experiments, indicating that the interactive ice sheet coupling has limited effect on the large-scale climate behaviour in our model under the given forcing. In particular, the evolution of the AMOC is hardly affected by the additional freshwater flux from GrIS mass loss in the coupled experiment (cf. Figure 7a and b), which amounts to 0.004 Sv, 0.052 Sv and 0.113 Sv averaged over the 21st, 22nd and 23rd century, respectively. The only global variable where differences are clearly visible is global sea surface salinity that is reduced in the coupled model compared to NorESM2fixed (Fig. 6d) in response to that additional freshwater input. A detailed analysis of the (regional) differences between the coupled NorESM2 and the version with fixed ice sheets NorESM2fixed can be found in Haubner et al. (in prep).


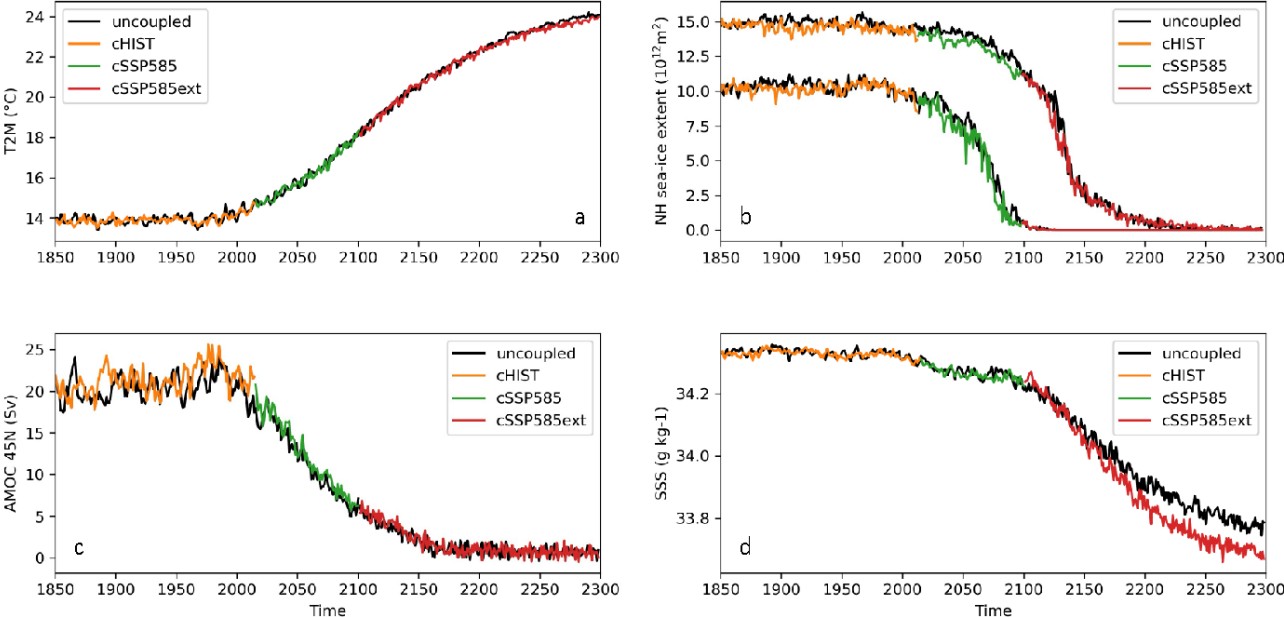

**Figure 6 Large-scale climate characteristics for NorESM2 (colour) compared to NorESM2fixed (black). a) 2-m air temperature, b) maximum and minimum northern hemisphere sea-ice extent, c) AMOC strength at 45°N, d) sea surface salinity.**






Increased mass loss of the GrIS compared to the historical background trend first emerges at the beginning
of the projection period ~2015-2025 (Fig. 7b). However, instead of further accelerating ice sheet retreat
after 2025, as might be expected from the global temperature evolution, we see a nearly constant rate of
mass loss until 2080. This can be explained by a rapidly weakening AMOC, which leads to regional
cooling in the North Atlantic that offsets a substantial part of the warming trend. Compared to results
based on standalone ice sheet simulations over the same period with a large range of models (Goelzer et
al., 2020), the projected sea-level contribution in NorESM2 is below the lower bound, which we attribute
to both the strong AMOC response and an initial cold bias of NorESM2. However, a similar experiment
with CESM2-CISM (Muntjewerf et al., 2020b) shows a strongly decreasing SMB already after 2050,
despite a decreasing AMOC, which may be explained by a different ocean model or different interdecadal
variability between the two global models.
Mass loss rate increases towards the end of the 21$^{st}$ century and continues to do so until the end of our
experiment in year 2300. The surface mass balance over the extension period is rapidly decreasing and
leads to a cumulated sea-level contribution of close to 1.5 m by 2300 (Figure 7a-b). The ice sheet loses
mass, thins by more than 1 km mainly around the coast, and exhibits retreat of several tens of km around
the entire margin (Figure 7c-d).

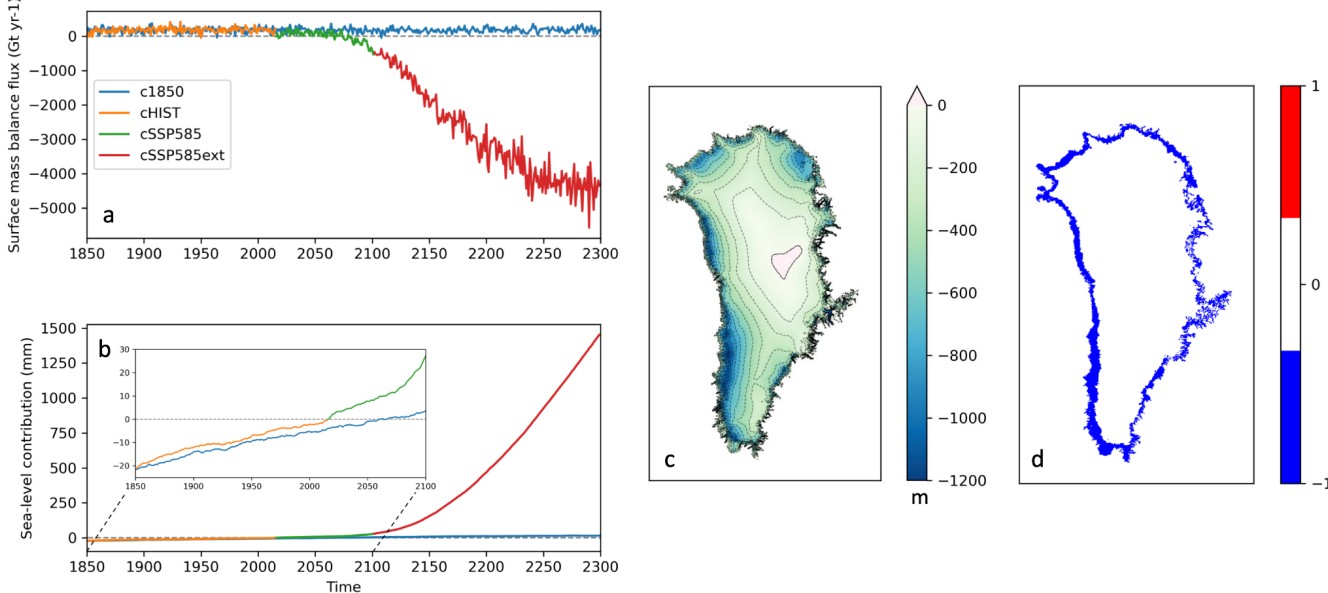

**Figure 7 GrIS characteristics for the chain of experiments: a) total surface mass balance, b) sea-level contribution, c) ice thickness**
**change and d) ice mask change between 2014 and 2300 (blue indicating retreat).**



## 5 Discussion

The presented model development and experiments represent the first interactive coupling of the GrIS in the global Earth System Model NorESM. This work shines light on challenges that are inherent to combining model components of different spatial resolution.

Climate model biases due to limited resolution of the atmospheric component are difficult to overcome, given that current global climate models are typically run at the upper limit of available High Performance Computing resources. While the elevation-class approach for downscaling is successful for SMB and surface energy components with strong temperature dependence, improving the distribution of mostly topographically controlled precipitation is very difficult. In this context, the potential of regional grid refinement is promising, a possibility that emerges with the CAM spectral element dynamical core (Van Kampenhout et al., 2019; Herrington et al., 2022) that will be available in future versions of NorESM.

NorESM2 has been initialised and run with a dynamic GrIS in a complementary way compared to the approach taken with CESM2 (Muntjewerf et al., 2020a; b). Compared to these studies, we have tried to initialise closer to observed ice sheet geometry and mitigate model drift by using stronger constraints on the ice sheet model. The approach of nudging the ice sheet thickness toward observed values during initialisation by calibrating the basal friction parameters is very effective but also has its caveats. Inaccuracies in model physics, parametrisations and boundary conditions are compounded into a modified basal friction field with effects that are hard to trace. In particular, any bias in the SMB (which we know can be substantial in some regions) is propagated into the dynamic behaviour of the ice sheet model in a non-transparent way (Berends et al. 2023). Masking the ice sheet to the observed present-day ice extent is also a strong limitation to the ice sheet physics and is only justified for the strong warming scenario applied here, where the entire ice sheet margin retreats. It remains a challenge to reduce the impact of climate and ice sheet biases (which are often mutually reinforcing) on the coupled state while maintaining the full prognostic capabilities of the model.

The assumption that the pre-industrial ice sheet state is close to the present-day observed one is questionable and could be refined e.g. by running one or several iterations from pre-industrial to 1990 with an updated 1850 state to better match the transient historical ice sheet state. However, such a refinement would add many more model years to the experimental setup, and its success could be dependent on controlling internal variability of the system. Reconstructions of the climate and ice sheet states further back in time, ideally towards the pre-industrial climate, would be very useful in this context. Since we have focused on describing the ice sheet coupling, we have not analysed the climate evolution over the historical and future period in great detail. A deeper analysis of differences between the coupled and uncoupled experiments can be found in a separate paper (Haubner et al., in prep). However, it is apparent that the influence of ice sheet changes on the global mean climate is rather limited in the current setup and for the given forcing. In particular, we may have expected a larger response of the AMOC to the additional freshwater input coming from Greenland, even if the lack of a dedicated ocean forcing in our setup may be under-estimating ice sheet retreat to some extent. It appears that the AMOC weakening in the model version without ice sheet coupling is already so intense in NorESM2 (Schwinger et al., 2022), that the freshening due to ice sheet meltwater fluxes has little additional effect.



## 6 Conclusions

This paper describes the first coupled climate–Greenland ice sheet model setup of NorESM and illustrates its behaviour with first simulation results. We have presented modelling choices which are effective in working around some of the climate biases and in preparing a present-day ice sheet state that is close to observations. The simulated present-day surface mass balance in NorESM captures the main features when compared to high-fidelity regional climate model simulations but does not represent the detailed distribution of precipitation very well due to the relatively coarse resolution of the atmosphere. Experiments under a strong future warming scenario until 2300 show a limited effect of including the Greenland coupling on most global variables under the given forcing.

Other challenges of coupling Earth system components of different typical response timescales and spatial resolution remain. Further work with NorESM is therefore ongoing e.g. to include the coupling of marine-terminating outlet glaciers with the ocean, and to improve the representation of SMB over the GrIS. We are also working towards coupling with the Antarctic ice sheet, which is an obvious next step but includes additional challenges, in particular a less effective downscaling of SMB boundary conditions due to a limited contribution of melt and the important interaction between ice shelves and the Southern Ocean.

## Code and data availability

The NorESM model code is developed and freely available under https://github.com/NorESMhub/NorESM. The specific code repository used to set up the model is archived under https://doi.org/10.5281/zenodo.11199967. The full code used to produce the coupled experiments is persistently archived under https://doi.org/10.5281/zenodo.11200059. The raw data for the coupled NorESM2 experiments has been archived with persistent identifiers https://doi.org/10.11582/2024.00079, https://doi.org/10.11582/2024.00080, https://doi.org/10.11582/2024.00081 and https://doi.org/10.11582/2024.00082. The CMORized output from the NorESM2fixed experiments (Bentsen et al., 2019a;b) can be accessed through the ESGF at https://doi.org/10.22033/ESGF/CMIP6.8040 and https://doi.org/10.22033/ESGF/CMIP6.8321.

## Author contributions

HG, PML and AB designed the experimental setup. HG developed the Greenland coupling with help of PML, AB, WHL, GL and KTC. HG conducted the coupled NorESM experiments and wrote the manuscript with help from all co-authors.

## Competing interests

The contact author has declared that none of the authors has any competing interests.



## Disclaimer

Publisher's note: Copernicus Publications remains neutral with regard to jurisdictional claims made in the text, published maps, institutional affiliations, or any other geographical representation in this paper. While Copernicus Publications makes every effort to include appropriate place names, the final responsibility lies with the authors.

## Acknowledgements

We thank Michael Schulz, Mats Bentsen,Trude Storelvmo and all the other KeyCLIM and INES project participants for discussions and suggestions that supported the model development and analysis of the simulations. We thank the Norwegian Climate Centre for providing NorESM2 data for CMIP6 and the Earth System Grid Federation (ESGF) for archiving the CMIP data and providing access. High-performance computing and storage resources were provided by Sigma2 - the National Infrastructure for High Performance Computing and Data Storage in Norway through projects NN9560K, NN9252K, NN2345K, NN8006K, NS9560K, NS9252K NS2345K, NS9034K and NS8006K.

## Financial support

This research has been supported by the Research Council of Norway under projects KeyClim (295046), INES (270061) and GREASE (324639). GL, WHL, and KTC are supported by the NSF National Center for Atmospheric Research, which is a major facility sponsored by the National Science Foundation under Cooperative Agreement no. 1852977.

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
