# Peer review of "Interactive coupling of a Greenland ice sheet model in NorESM2 1"

_EGUsphere, 2024_

## Referee Comment (RC2)

*Review of Goelzer, Langebroek, Born, Hofer, Haubner, Petrini, Leguy, Lipscomb, and Thayer-Calder: "Interactive coupling of a Greenland ice sheet model in NorESM2." (GMD, Paper:* egusphere-2024-3045*)*

The manuscript of Goelzer and others describes the coupling between the Norwegian Earth System Model (NorESM) and the Community Ice Sheet Model (CISM), where the latter represents the Greenland ice sheet. The focus of the paper is the implementation of the coupling, a short description of the initialization, and a brief analysis of performed simulations starting in 1850 and ending in the year 2300, focusing on the future warming scenario following an extended SSP5-8.5 scenario. They briefly show the influence of the ice sheet interaction on the climate and the ice sheet evolution. I'm particularly intrigued by the related paper in preparation (Haubner et al.) and look forward to a more in-depth analysis.

It was an absolute pleasure to read the well-structured and prepared manuscript. The figures are of good quality, necessary, and informative. This work is highly relevant given the number of groups working on the forefront topic of the interaction between Earth System models and dynamical ice sheet models. This work is also intriguing for the general ice sheet modeling community and those using ice sheet model projections to determine the future sea level.

**I recommend the publication of the manuscript after minor corrections.**

**General comments**

The manuscript is well-organized and written.

In the coupled model, the Greenland ice sheet has a limited impact on global climatic conditions if we ignore the contribution of a declining ice sheet to the sea level. Is it characteristic of the used model system, or would the authors generalize these results? If we turn towards the sea level response, do the authors detect different sea level responses between simulations where CESM interacts with NorESM and where it "only" receives the forcing from an uncoupled NorESM simulation – without providing feedback? The latter would be comparable with typical ISMIP standalone simulations.

I'm unsure about GMD's standards, but you may please check the consistent use of "e.g." versus "e.g.," as well as " … and …" versus "…, and …" in lists.

**Specific comments**

Main document

Page 3, Line 97 (P3, L97): Since you are using an unequal on a variable-thickness sigma coordinate system, I wanted to ask if you have limited the vertical resolution of the lowest layer to avoid numerical issues. Furthermore, what is the typical and minimum lowest layer thickness?

P3, L99: I'm confused about the conflicting information about a domain on a polar stereographic projection while the horizontal resolution shall be 4 km (e.g., P4, L21). Please clarify.

P4, L15: Does the SMB calculation allow for positive and negative sublimation?

P4, L33: What justifies the uniform lapse rate of -6 K km$^{-1}$? Please also check the provided unit "°K/km," which might contain a mixture of Degrees Celsius and Kelvin.

P4, L41–42: You write, "… for lower snowpack depth, the accumulated snow does not directly contribute to the SMB." What is meant by direct contribution, and what would be an indirect contribution?

Please clarify.

P4, L44–45: You mention "a horizontal bilinear interpolation and a linear vertical interpolation between" the models. It raises the question of whether the order of these interpolations influences the results. If so, how big is the difference?

P5, L47: In the model setup, masks are used "for the accumulation region, and one for the ablation region." It appears these are computed for each year. Are these masks changing within a year? Please clarify.

P5, L67: You "update the topography every five years" in your simulations. I understand that the commonly small changes justify these five years. Nevertheless, would it be better to tricker updating the topography once an orography change exceeds a given threshold, e.g., $|\Delta h(x,y)| > h_{threshold}$?

P5, L77–78: Do the authors mean "Solid ice fluxes are cumulated and passed to the ocean annually" as a mean flux?

P9, L59: What are the starting carbon dioxide emissions in the year 2300 when the emission starts to decline linearly? Therefore, the authors may write: "... reduced linearly starting from XX Gt year$^{-1}$ in 2100 to … ."

P10, L80–81, L82, L84: You may help the reader to link the provided information in the text to the related subfigure. For instance, the authors may write:"… directly by NorESM2-EC (Fig. 4a, NorESM ...) compared ... (Fig. 4b, NorESM2-MAR, Fettweis et al., 2017). This … for the same period (Fig. 4C, ERA5-MAR),… ."

P13, Figure 7c: If I understand correctly, a changing ocean forcing is not implemented. Therefore, what drives the elevation reduction in northeast Greenland at the mouth of the 79-glacier (nioghalvfjerds)? Might it be related to the tuning of the ice sheet's basal conditions?

P14, L94–95: I'm with you that "Reconstructions of the climate and ice sheet states further back in time … would be very useful in this context." Is the work of Kjær et al. (2012) and Bjørk et al. (2012) relevant in this respect? Could the authors please provide information on what they would like to obtain from the community?

P14, L104: Is the result that "freshening due to ice sheet meltwater fluxes has little additional effect" on the AMOC a consequence of the analysis of Mikolajewicz and Maier-Reimer (1994)?

P16, L65: Please check the indent of citations.

Figure

Figure 1: What do the arrows represent? Please clarify. Also, the authors should consider skipping all arrows that are not active instead of having one disabled arrow, e.g., the strikethrough arrow.

Figure 2: In subfigure c), the color bar indicates that the value range does not exceed surface elevation differences of ±250 m. If not, please adjust the colorbar or mention it in the figure caption. Also, check this issue for the remaining figures, e.g., four (4) and seven (7).

Figure 4: Since the caption says that all "fields are masked to the modeled ice sheet area in NorESM2 at the end of the year 2014," it raises the question of whether the ice sheet has retreated in some areas. If so, please indicate lost ice.

Figure 5: What does the gray-shaded area mark? Please clarify it in the figure caption.

Figure 6: In the figure caption, please add the information that the air temperature and surface salinity

are global, e.g., "a) global 2-m air temperature" and " d) global sea surface salinity."

**Bibliography**

Bjørk, A. A., Kjær, K. H., Korsgaard, N. J., Khan, S. A., Kjeldsen, K. K., Andresen, C. S., Box, J. E., Larsen, N. K., & Funder, S. (2012). An aerial view of 80 years of climate-related glacier fluctuations in southeast Greenland. Nature Geoscience, 5, 427–432. https://doi.org/10.1038/ngeo1481

Kjær, K. H., Khan, S. A., Korsgaard, N. J., Wahr, J., Bamber, J. L., Hurkmans, R., van den Broeke, M., Timm, L. H., Kjeldsen, K. K., Bjork, A. A., Larsen, N. K., Jorgensen, L. T., Faerch-Jensen, A., & Willerslev, E. (2012). Aerial Photographs Reveal Late-20th-Century Dynamic Ice Loss in Northwestern Greenland. Science, 337(6094), 569–573. https://doi.org/10.1126/science.1220614

Mikolajewicz, U., & Maier-Reimer, E. (1994). Mixed boundary conditions in ocean general circulation models and their influence on the stability of the model's conveyor belt. Journal of Geophysical Research, 99(C11), 22633–22644. https://doi.org/10.1029/94JC01989

---

## Author Response (AR1)

**Authors' reply to reviewer comments**

**Reply to comments by Anonymous Referee #1**

We would like to thank Referee #1 for the good suggestions that will help to improve our manuscript. We are confident that we can address the comments (**bold**) in a revised version. Below we provide a point-by-point response to the comments.

Goelzer et al. present the inclusion of a dynamic ice sheet model (namely CISM) into the NorESM2 general circulation model. They describe the methodology of the coupling and their initialisation procedure. They finally show coupled ice sheet – climate model experiments for the historical period and into the future, until 2300. The paper is very clear and nicely written. However, it feels a bit short and does not provide much comparison with previous works and does not provide a lot of material to fully grasp the limit of the approach chosen. Also the main finding is that the interactive ice sheet does not matter much but in fact this been looked with very simple integrated metrics where it would have been useful to discuss spatial patterns of Northern Hemisphere climate change induced by the ice sheet retreat. All this information would be useful for other groups aiming at doing a similar work with alternative climate / ice sheet models. More details in the following.

Thank you very much for the positive evaluation and for providing this perspective. We agree that adding some more material can improve the usefulness of the paper. We have added new figures in the main manuscript and in the supplement.

**Major comments**

- Discussion with respect to CESM2-CISM previous coupling. The authors acknowledge that they share various important components and methodologies with the CESM2-CISM model. To my understanding, the coupling strategy is identical and only the initialisation strategy is different. Is this correct?

Yes, the coupling strategy is the same as in CESM2-CISM. We call it "our novel coupled modelling framework" (see below) because for NorESM this is the first time the ice sheet is included.

If yes I think it should be clarified stronger to put more weigh on the things that are really different. At the moment it is not very clear for example if the elevation class methodology shows subtle differences with Muntjewerf et al. (2021) or if it is exactly the same. P2L70 for example we can read "we describe our novel coupled modelling framework" but then, reading the text I cannot find any difference with the CESM2-CISM coupling. I understand that CESM2 and NorESM2 are two different models, with different oceanic components and different climate sensitivities, so I support the paper. But I think it would be more useful to stress on what you have kept, what you have not, and why.

Agreed. We have provided a clearer distinction between NorESM2 and CESM2 and have put more focus on the differences between them in the revised manuscript. The elevation class methodology is the same as in CESM2 as was already stated in the manuscript.

- Discussion on the initialisation strategy. I understand that it was a practical choice to impose an ice sheet mask and perpetual calving for floating points in order to maintain a good agreement with respect to present-day observed ice sheet topography. This point is clearly explained in the methods and it is shortly discussed in the discussion section. However I think it deserves further attention. As the authors acknowledge, the climate model can present large atmospheric biases (see also my next comment) and these biases are somehow compensated by ice sheet dynamics. The way I see it is that an overall overestimation of SMB (particularly true for North and East of the ice sheet) will translate in reduced basal drag in order to push the excess ice outside the observed ice mask, where this excess ice is simply removed independently from the simulated SMB / oceanic characteristics. It is possible that the "good" match with present-day topography hides efficiently the cold bias of NorESM. In turns, it is also possible that this limits the ice sheet model response to future warming. I wonder if it would have been possible to use some king of bias correction to evaluate this? Even a simple one? Right now I do not find the coupling very convincing since it relies heavily on this "masking" method. Again, I understand why it has been done this way, but at least I expect to see clear and documented justification of it. For example even a control (continuation of pre-industrial) coupled simulation without any masking to show the unwanted ice extension would be useful. I think this problem is common to every modelling groups and I think it is nice to document how far we are and what we can do and we cannot.

Thank you for the suggestions. We agree that the chosen method is a practical workaround for the climatic biases, and we have now illustrated that better with additional experiments and figures. We have provided further arguments and a better justification for the initialisation approach.

In places where the SMB is positive outside of the observed ice sheet mask, we know that the ice sheet will just expand and grow out towards the land-sea mask (and in fact beyond, if not controlled there). Therefore, we did not consider running additional coupled experiments without masking very useful. We have illustrated the issue instead with standalone ice sheet simulations forced by NorESM2 output.

- Presentation of model biases. It would be useful to present the climate model biases in terms of regional temperature (annual and summer), precipitation, but perhaps also short/long wave etc. At present there is only a comparison with MAR SMB, restricted over the ice sheet but it would be interesting to also know what happens outside the ice sheet (tundra).

Agreed. We have included further analysis and figures that go beyond the ice sheet mask.

- Impact of interactive ice sheets on the simulated future climate. It is a bit overlooked I think since only simple integrated metrics are shown. I think that a map of regional temperature change w/o the coupling would be useful for instance.

There is a companion paper now under revision and out in discussion in ESD (Haubner et al., https://doi.org/10.5194/egusphere-2024-3785) with specific focus on the comparison between coupled and uncoupled future experiments. We have made the link to this paper stronger, but want to leave room for the more detailed discussion in that paper.

**Minor comments / questions**

P2L50. Climate biases and climate downscaling are two different problems. We can do a very neat downscaling but if the biases are strong we will not have a high-quality surface mass balance.

OK, has been reformulated.

**P4L15. Why not total precipitation instead? Rain is also contributing to SMB via refreezing.**

That is why there is a refreezing term.

**P4L33. Why this value? Why not the model vertical lapse rate? Do you have runs with different values of this and check its importance on SMB?**

We would have liked to experiment with this, but for now this is hard-coded in the land model and applies globally. This is something we want to explore in NorESM3.

P4L35. Do you have a justification for no-change in relative humidity? It seems not really driven by observations... Again, do you have experiments where you impose a vertical gradient for humidity (specific may be more appropriate). Perhaps it would have made more sense to me to keep the same specific humidity in the vertical but recomputing the relative humidity (hence producing precipitation eventually).

This approach is the standard for glaciated regions in the land model and we have not explored other possibilities. Keeping relative humidity constant in the vertical implies that the downscaled specific humidity does change depending on temperature, which is the desired effect. This means e.g. that air with 100% relative humidity at a given mean elevation h0 will 'lose' water content when downscaled to a higher elevation h1>h0 to maintain relative humanity constant at 100%. We believe this is in line with the expectation of the referee. In addition, there is a partitioning of precipitation based on the downscaled temperature, allowing rain to fall at lower elevations while snow falls at higher elevations.

**P5L67. It might useful to run some additional sensitivity tests with different update frequencies. Is there a limit from which we observe unwanted jumps for some climatic variables?**

Because of the relatively coarse resolution of the atmosphere, the 5-year update interval is not close to any critical threshold in our case. But optimising this (i.e. choosing a less frequent

update) will depend on the rate of elevation change and therefore on the climate scenario. We have added a user recommendation along those lines.

P5L60-69. Do the model includes some kind of ice mask where the albedo cannot reach too low values. Have you tried to separate the effects of changing the ice mask and changing the albedo for the future simulations?

Good point. The ice mask (and the albedo) is changing dynamically with the modelled ice cover. This information has been added to the section before entitled "Surface energy balance and surface mass balance". We have not attempted to change the albedo parameterisation. See also response to comment **P10 Figure 4** below.

**P5L78. Homogeneously distributed within the year?**

Yes. Thanks, has be added.

P7 Figure 2. The agreement is really nice. Can you provide an estimate of the drift for this? You could show the map of thickness difference with respect to observations at the end of the cControl experiment.

The drift under constant forcing can generally be very small. Here we have chosen to impose a slight background trend as shown in 3d. We have added figure S4 to show how the difference to the observed surface elevation has changed towards the end of the historical simulation.

P7 Figure 2. In the supplement you could show the inferred basal drag coefficient with some details on the law you used. Also a map of observed velocities with respect to simulated ones will be useful, since the climate model biases are compensated by ice dynamics.

OK, figures of the inferred bed stress coefficient beta and velocity have been added to the supplement. as suggested.

**P9 Figure 3. Temperature and precip: is it a spatial average? Over which domain?**

Yes, global means. Has been added.

P10 Figure 4. I imagine that the snow albedo can be tuned somehow in the model. Can you provide some info on how it is computed? Have you tried to tune it? How good is it with respect to satellite observations, and/or MAR?

We have added information on how albedo is calculated in 2.3. We didn't do any (additional) tuning and did not compare it against observations or other products. However, a lot of work on CLM has been done in earlier work with CESM (notably https://doi.org/10.1029/2019JF005318). This reference has been added.

P12L36. Does NorESM2 includes some freshwater flux scenarios when performing future projections? Do you still have these when you activate the coupling? How large is the scenario compared to your estimated fluxes?

No, we did not prescribe any freshwater flux scenarios. Freshwater fluxes are physically produced in NorESM2, whether coupled or not.

P12 Figure 6. T2m, SSS: global average?

Yes, indeed. Has been added.

P12L36. Since you have a strong change in AMOC I think it would be useful to have more discussion on the freshwater flux and their impact. Do you some different spatial structures with/without the coupling when looking at the North Atlantic surface and subsurface temperatures? If yes do they come from topography/ice mask changes or freshwater flux?

This is discussed in some detail in Haubner et al., which shows SST cooling around Greenland and warming of the GrIS margin due to the lapse-rate effect. We have made a close link to that companion paper in the text.

P13L49. I think it is not enough to simply mention the paper in preparation. Since AMOC is the only change you have with respect to the standard uncoupled simulation, it would be nice to have some plots of regional oceanic changes.

The Haubner et al. paper is now under review and in open discussion. We try to avoid too much overlap with that paper.

P13 Figure 7. c1850: should be cControl instead?

Correct, Thanks for spotting this.

P14L04. But this might simply results from the limited ice sheet response in these simulations.

I would agree for the  $21^{st}$  and  $22^{nd}$  century. It takes some time to get going, but over the  $23^{rd}$  century, the GrIS loses ~ 1m SLE with a freshwater flux > 0.1 Sv. That is not a limited ice sheet response.

**Technical corrections**

P1L33. Surface albedo is meant?

OK, corrected.

P12L40. Not useful to cite this since there is no preprint available.

We had included it, because it is an important companion paper. The preprint is now available and the paper is being revised in parallel.

**Reply to comments by Anonymous Referee #2**

We would like to thank Referee #2 for the thorough and valuable comments. We appreciate the opportunity to address the comments (**bold**) in a revised version. Below we provide a point-by-point response to the comments.

The manuscript of Goelzer and others describes the coupling between the Norwegian Earth System Model (NorESM) and the Community Ice Sheet Model (CISM), where the latter represents the Greenland ice sheet. The focus of the paper is the implementation of the coupling, a short description of the initialization, and a brief analysis of performed simulations starting in 1850 and ending in the year 2300, focusing on the future warming scenario following an extended SSP5-8.5 scenario. They briefly show the influence of the ice sheet interaction on the climate and the ice sheet evolution. I'm particularly intrigued by the related paper in preparation (Haubner et al.) and look forward to a more in-depth analysis.

It was an absolute pleasure to read the well-structured and prepared manuscript. The figures are of good quality, necessary, and informative. This work is highly relevant given the number of groups working on the forefront topic of the interaction between Earth System models and dynamical ice sheet models. This work is also intriguing for the general ice sheet modeling community and those using ice sheet model projections to determine the future sea level.

I recommend the publication of the manuscript after minor corrections.

Thank you very much for the positive evaluation.

**General comments**

The manuscript is well-organized and written.

Thanks.

In the coupled model, the Greenland ice sheet has a limited impact on global climatic conditions if we ignore the contribution of a declining ice sheet to the sea level. Is it characteristic of the used model system, or would the authors generalize these results? If we turn towards the sea level response, do the authors detect different sea level responses between simulations where CESM interacts with NorESM and where it "only" receives the forcing from an uncoupled NorESM simulation – without providing feedback? The latter would be comparable with typical ISMIP standalone simulations.

These are interesting questions. Concerning the first point, we are careful not to generalise, since it is difficult to predict how other ESMs will respond. Instead, we are looking forward to seeing similar experiments with other models in the future, e.g. as organised under ISMIP7, which will allow for a proper comparison. This is now mentioned in the discussion.

In the presented runs, we are unfortunately not able to get ice sheet forcing from the uncoupled NorESM2, as part of the simulation was run without the relevant output. This, we feel, renders the response below a bit too speculative to present as discussion in the paper but we provide it here for reference. In order to better answer similar questions in the future, the possibility to work with different data models is an absolute priority in our ongoing development work and an early version of NorESM3 already includes those options, which we will emphasize. Based on the analysis in Haubner et al., we speculate that the difference by 2100 between the coupled NorESM2-CISM and a standalone CISM forced with NorESM2 output would be minor. By 2300 there is a strong ocean cooling around the GrIS in NorESM2-CISM in response to anomalous freshwater runoff, which should mitigate some of the mass loss. Conversely, we may expect a NorESM-forced standalone experiment to overestimate the sea-level contribution. But this is under the assumption that the documented temperature change in NorESM2-CISM is only due to a local lapse-rate effect and that it would be represented in the standalone experiment in a similar way (with the elevation class approach). In any case, please note that the results would be very different from the ISMIP6 experiments where the climate forcing was regionally downscaled and bias corrected by using an anomaly forcing approach.

For what it is worth, when running standalone CISM forced with the output of the NorESM2-CISM coupled simulation, it produces the exact same sea-level contribution (and general ice sheet state) as the coupled model.

I'm unsure about GMD's standards, but you may please check the consistent use of "e.g." versus "e.g.," as well as " ... and ..." versus "..., and ..." in lists.

Thanks. We are not using Oxford spelling nor Oxford (serial) commas. See <a href="https://www.geoscientific-model-development.net/submission.html#english">https://www.geoscientific-model-development.net/submission.html#english</a>

**Specific comments**

**Main document**

Page 3, Line 97 (P3, L97): Since you are using an unequal on a variable-thickness sigma coordinate system, I wanted to ask if you have limited the vertical resolution of the lowest layer to avoid numerical issues. Furthermore, what is the typical and minimum lowest layer thickness?

The surface layer is 23% and the lowest layer is 3.6% of the ice column everywhere, with the absolute thickness (in meters) depending on the local ice thickness. We will add this information. In our experience there are no numerical issues with this scheme. We found that running with 11 vertical layers is a good compromise in optimizing the computational cost of the simulation versus numerical simulation accuracy. Using the DIVA solver, adding more vertical layers has a limited impact on the ice sheet dynamic representation.

**P3, L99: I'm confused about the conflicting information about a domain on a polar stereographic projection while the horizontal resolution shall be 4 km (e.g., P4, L21). Please clarify.**

We are now using the same notation ("4X4 km horizontal resolution") in both places. Otherwise, it is not clear to us what the conflicting information is. Maybe Referee #2 means that the stereographic projection does not preserve area and the resolution is approximately around 4x4km, but not exact? It is common the specify grid resolution in km, even if the area is slightly varying across the domain.

**P4, L15: Does the SMB calculation allow for positive and negative sublimation?**

Sublimation is always positive. So the process is removing mass. This has been added.

**P4, L33: What justifies the uniform lapse rate of -6 K km-1? Please also check the provided unit "oK/km," which might contain a mixture of Degrees Celsius and Kelvin.**

See similar comment by Referee #1. We would have liked to experiment with this, but for now this is hard-coded in the land model and applies globally. This is something we want to explore in NorESM3. Corrected units to "oC/km".

**P4, L41–42: You write, "... for lower snowpack depth, the accumulated snow does not directly contribute to the SMB." What is meant by direct contribution, and what would be an indirect contribution? Please clarify.**

This is a question of shared responsibility between ice sheet and land model. Accumulation below 10 m is handled by the land model without impact on ice sheet mass budget. We have reformulated this sentence.

**P4, L44–45: You mention "a horizontal bilinear interpolation and a linear vertical interpolation between" the models. It raises the question of whether the order of these interpolations influences the results. If so, how big is the difference?**

The order of the two interpolations is not an option and reversing it would not make sense. The horizontal interpolation produces forcing on each ice sheet grid point location, which is then corrected (interpolated between elevation classes) to the specific, high-resolution surface elevation.

**P5, L47: In the model setup, masks are used "for the accumulation region, and one for the ablation region." It appears these are computed for each year. Are these masks changing within a year? Please clarify.**

No. The SMB is produced every full year. The masks and the normalisation are also updated every full year. Added clarification.

P5, L67: You "update the topography every five years" in your simulations. I understand that the commonly small changes justify these five years. Nevertheless, would it be better to tricker updating the topography once an orography change exceeds a given threshold, e.g.,  $|\Delta h(x,y)| > h$ threshold?

See also reply to similar comment from Referee #1. Since the procedure is not implemented in model memory, it is more practical to update at a fixed frequency. Here the restart frequency of the model.

P5, L77–78: Do the authors mean "Solid ice fluxes are cumulated and passed to the ocean annually" as a mean flux?

Yes, thanks. This was also remarked by Reviewer #1.

P9, L59: What are the starting carbon dioxide emissions in the year 2300 when the emission starts to decline linearly? Therefore, the authors may write: "... reduced linearly starting from XX Gt year-1 in 2100 to ...."

The fossil  $CO_2$  emissions are  $\sim 35$  Gt year-1 in 2100. Text has been updated as suggested. Thanks.

P10, L80–81, L82, L84: You may help the reader to link the provided information in the text to the related subfigure. For instance, the authors may write:"... directly by NorESM2-EC (Fig. 4a, NorESM ...) compared ... (Fig. 4b, NorESM2-MAR, Fettweis et al., 2017). This ... for the same period (Fig. 4C, ERA5-MAR),...."

OK. Text has been updated.

P13, Figure 7c: If I understand correctly, a changing ocean forcing is not implemented. Therefore, what drives the elevation reduction in northeast Greenland at the mouth of the 79-glacier (nioghalvfjerds)? Might it be related to the tuning of the ice sheet's basal conditions?

The driver is mostly decreasing SMB as discussed in the text and as can also be seen in Figure 7. of Haubner et al. https://doi.org/10.5194/egusphere-2024-3785.

P14, L94–95: I'm with you that "Reconstructions of the climate and ice sheet states further back in time ... would be very useful in this context." Is the work of Kjær et al. (2012) and Bjørk et al. (2012) relevant in this respect? Could the authors please provide information on what they would like to obtain from the community?

Thanks. Those are very relevant references. We have included those as examples in the discussion.

P14, L104: Is the result that "freshening due to ice sheet meltwater fluxes has little additional effect" on the AMOC a consequence of the analysis of Mikolajewicz and Maier-Reimer (1994)?

No, that is not the case.

P16, L65: Please check the indent of citations.

OK. Corrected.

**Figure**

Figure 1: What do the arrows represent? Please clarify. Also, the authors should consider skipping all arrows that are not active instead of having one disabled arrow, e.g., the strikethrough arrow.

The arrows represent two-way or one-way interactions. This has been clarified in the figure caption. We prefer to specifically show the missing interaction, which is physically there, but not implemented in the model.

Figure 2: In subfigure c), the color bar indicates that the value range does not exceed surface elevation differences of  $\pm 250$  m. If not, please adjust the colorbar or mention it in the figure caption. Also, check this issue for the remaining figures, e.g., four (4) and seven (7).

OK, will be done.

Figure 4: Since the caption says that all "fields are masked to the modeled ice sheet area in NorESM2 at the end of the year 2014," it raises the question of whether the ice sheet has retreated in some areas. If so, please indicate lost ice.

This figure is focussed on comparing SMB and does so with a common ice mask for clarity. We do not want to change that. In any case, the ice mask does change ever so slightly in line with the 20 mm ice mass loss over the historical period. It would hardly be visible on a map view, though. We are not convinced this would add relevant information.

Figure 5: What does the gray-shaded area mark? Please clarify it in the figure caption.

This is the period over which "SMB trends after 1990" are calculated. Has been added to the caption.

Figure 6: In the figure caption, please add the information that the air temperature and surface salinity are global, e.g., "a) global 2-m air temperature" and "d) global sea surface salinity."

**Bibliography**

Bjørk, A. A., Kjær, K. H., Korsgaard, N. J., Khan, S. A., Kjeldsen, K. K., Andresen, C. S., Box, J. E., Larsen, N. K., & Funder, S. (2012). An aerial view of 80 years of climate-related glacier fluctuations in southeast Greenland. Nature Geoscience, 5, 427–432. https://doi.org/10.1038/ngeo1481

Kjær, K. H., Khan, S. A., Korsgaard, N. J., Wahr, J., Bamber, J. L., Hurkmans, R., van den Broeke, M., Timm, L. H., Kjeldsen, K. K., Bjork, A. A., Larsen, N. K., Jorgensen, L. T., Faerch-Jensen, A., & Willerslev, E. (2012). Aerial Photographs Reveal Late-20th-Century Dynamic Ice Loss in Northwestern Greenland. Science, 337(6094), 569–573. https://doi.org/10.1126/science.1220614

Mikolajewicz, U., & Maier-Reimer, E. (1994). Mixed boundary conditions in ocean general circulation models and their influence on the stability of the model's conveyor belt. Journal of Geophysical Research, 99(C11), 22633–22644. https://doi.org/10.1029/94JC01989

Haubner, K., Goelzer, H., and Born, A.: Limited global effect of climate-Greenland ice sheet coupling in NorESM2 under a high-emission scenario, EGUsphere, 2025, 1–25, <a href="https://doi.org/10.5194/egusphere-2024-3785">https://doi.org/10.5194/egusphere-2024-3785</a>, 2025.